# Psychometric Validation of the Adult Prosocialness Behavior Scale in a Professional Teaching Context

**DOI:** 10.3390/bs13090761

**Published:** 2023-09-13

**Authors:** Manuel Mieres-Chacaltana, Sonia Salvo-Garrido, Sergio Dominguez-Lara, José Luis Gálvez-Nieto, Paula Alarcón-Bañares

**Affiliations:** 1Departamento de Diversidad y Educación Intercultural, Universidad Católica de Temuco, Temuco 4780000, Chile; mieres@uct.cl; 2Departamento de Matemática y Estadística, Universidad de La Frontera, Temuco 4780000, Chile; 3South American Center for Education and Research in Public Health, Universidad Privada Norbert Wiener, Lima 15108, Peru; sergio.dominguez@uwiener.edu.pe; 4Departamento de Trabajo Social, Universidad de La Frontera, Temuco 4780000, Chile; jose.galvez@ufrontera.cl; 5Departamento de Psicología, Universidad de La Frontera, Temuco 4780000, Chile; paula.alarcon@ufrontera.cl

**Keywords:** prosociality, teaching, teachers, education, adulthood, psychometric validation, measurement invariance

## Abstract

For the teaching profession, prosociality is a relevant skill. From this perspective, the present study aimed to examine the psychometric properties of the Adult Prosocialness Behavior Scale (APBS) in a professional setting of primary school Chilean teachers (*n* = 1404; mean age = 41.4; SD = 10.8; 77.4% women). ESEM and CFA were applied to evaluate its factor structure, internal consistency, factorial reliability, and invariance. The results supported a bifactor ESEM model in which prosociality was represented by a general latent factor and four specific factors (helping, sharing, caring, and empathy). A predominance of the unidimensional component of the APBS was observed (general factor, ECV = 0.938; ωh = 0.945), with adequate reliability (α = 0.932; ω of the factor general = 0.968) and invariance of the measurement in terms of gender. Its adequacy was confirmed by a good level of fit (RMSEA = 0.042 90%CI [0.036–0.049]; SRMR = 0.012; CFI = 0.995; TLI = 0.988). It is concluded that the APBS is a suitable instrument to measure prosociality in the professional setting studied. Its general and specific dimensions are relevant to the prosocial behaviors currently required of teachers. Future studies could contribute evidence of the transcultural validation of the APBS in other educational contexts in order to undertake comparative studies.

## 1. Introduction

The study of prosociality, generically defined as “dispositions, voluntary behaviors and processes that focus on or contribute to the well-being of others” on different levels [1] (p. 9), is of interest and relevance to educational research. Promoting prosocial behaviors in 21st-century schools is decisive in developing the bases of coexistence and a harmonious learning environment [2,3]. Its impact on peace and the prevention of bullying and violence in the school itself [4] is projected onto society [5]. It is positively related to social support and resilience [6] and negatively to criminal behavior [7]. It contributes to the regulation of emotions and increase in empathy [8], the reduction in discriminatory, racist, and aggressive attitudes and acts [9], and the advancement of historically marginalized groups [10]. In addition, a school climate characterized by support, care, and collaboration is fundamental to learning [11,12,13] as it promotes socially shared representations of its associated processes and better metacognitive adjustments [14]. At the same time, it is an important contextual factor for the development of character in children [15], as well as their motivation [16], well-being [17], satisfaction with life [18], and prosocial development, particularly among those from lower social classes [19].

The creation of environments conducive to learning depends largely on the talent and self-efficacy of teachers [20]. The deployment of their social and emotional competencies improves classroom relationships [21,22] and increases students’ prosocial motivation as they are role models for them [23]. In addition, it fosters cognitive activation [24], participation [25], autonomy [26], and academic self-efficacy [27]. On the other hand, its agency can induce collective efficacy to reduce aggression in school [7,28,29] and become a vigorous support for students who have suffered adversity [30], trauma [31], and addiction problems [32]. The benefit extends to the teachers themselves since greater prosociality in the classroom is positively associated with greater commitment [33] and job satisfaction [34] at a time when conditions of increased job stress pose short- and long-term threats to the supply of teachers [35].

Among the prosocial teaching behaviors reported in the literature, there are different types of aid, such as the relational type [36] and the help deployed for students with behavioral problems [37]; both are influential in improving school coexistence. Another type of aid is the action of sharing. Generous teachers can serve as role models and encourage children to take actions aimed at sharing and giving to others [38]. In addition, among the contributors to academic and social success are the caring actions practiced at school in the context of teacher–student relationships [39,40]. Direct and proactive actions intentionally expressing caring and trust [41] require teachers’ sensitivity [42].

At the same time, the dimension of empathy emerges, which is fundamental to prosociality since the person needs, first, to recognize and understand cognitively and emotionally what the other is feeling [32]. In this sense, empathy is a predictor of prosociality [7]. Teacher empathy has been defined as a teacher’s ability to include and share the emotional states of their students [43], and its relevance as an attribute has been demonstrated [44]. Greater empathy is related to a better preparation to address bullying [45], a greater motivation to display fair behaviors with the students [46], and a greater development of these aspects in the long term [47]. It can contribute to an increase in intergroup contact and a reduction in conflicts and discrimination in settings of ethnic and cultural diversity [48]. Emotional intelligence is a predictor of prosocial behaviors [49]. In short, teachers’ socioemotional competences are essential to education quality [49] and the optimization of student achievement [50].

In the case of Chile, there are regulatory provisions related to the prosocial behavior required of teachers which coincide with those reported in the previous paragraphs. For example, they help students address multiple cognitive, procedural, and attitudinal challenges [51] (p. 48). Additionally, they provide a welcoming and mutually caring environment [51]. Actions aimed at sharing practices and knowledge among teachers are added, as collaborative work is recognized as a fundamental axis of the profession [52]. Of course, all this shifts to the classroom with actions designed to share learning and knowledge collaboratively [51] (p. 49). Finally, they foster social practices in students such as empathy and listening to their interactions with peers and adults [51] (p. 41).

Regarding prosociality and its primary trait as voluntary behavior oriented to benefit others, the discussion has focused on its specific dimensions, contexts, and predictors. The diversity of prosocial behaviors and underlying motivations has made a consistent understanding of the construct complex [53], although a singular and unidimensional concept has predominated [54]. However, even though this general characteristic prevails, it is important to consider the different types of prosociality [55] and their operationalization in specific organizational contexts [56]. On the other hand, the differences noted with respect to gender [57], particularly the emotional support [58] and empathy [59,60] commonly favorable to women, appear to be associated with cultural stereotypes [61], which are especially influential during adolescence [62]. It is usually expected that women are more sensitive, empathic, and prosocial [63]; these are expectations that are likely linked to socialization practices [64], which would be reproduced at school [61] with consequences for student performance [65]. Finally, concerning the research on prosociality in different stages of life, the current state of research makes it possible to confirm that it is less understood in adolescence [66,67] and adulthood [68]. In concomitance with this, the construction of measurement instruments has been relatively scarce for this last stage [69,70]. And, generally, the application of advanced methods of psychometric analysis to the validity studies of prosocial behavior is still an emerging practice [71].

Consequently, prosociality is of critical importance to the teaching profession. Nowadays, both its behavioral and emotional dimensions are essential attributes for those who work in the teaching profession. Hence, there is a need to construct [72] or evaluate the psychometric properties of the measuring instruments in this professional area in which prosociality becomes a relevant job skill [73]. The present work is positioned within this second option. The Adult Prosocialness Behavior Scale (APBS), developed in Italy by Caprara et al. [74], is used to measure an adult’s prosociality [75]. It conceives it as a set of behaviors and feelings reflected in four types of actions: “helping, sharing, caring, and feeling empathy with others” [74] (p. 78). By including empathy—a controversial situation for other studies—the authors have substantiated that “in adulthood, one’s empathic motives or predispositions are not merely a correlate of his or her tendency to act prosocially but, rather, an integral part of such a tendency” [74] (p. 80). In this light, what are the factor structure, internal consistency, reliability, and factorial invariance of the APBS in a professional teaching context? The present study aims to assess these aspects with respect to the APBS in a sample of Chilean teachers.

In relation to the specific professional sphere being studied, the APBS has been evaluated with Chilean university students in teaching programs [76] and with teachers in state schools in Turkey [34]. However, those studies did not include an analysis of invariance by gender, and the former did not include teachers. Regarding the absence of the analysis of invariance, this could represent a limitation as it would assume that the measurement is equivalent between men and women, even when the evidence suggests that prosocial behavior develops differently due to the type of socialization: women are more oriented to caring, whereas men are instructed not to show such behaviors [77]. In the same way, it is important to have the perspective of in-service teachers since they have more experience and maturity than pre-service teachers [78]. In contrast, student teachers share pre-professional internship time with classes associated with their professional training.

From a general perspective, a recent meta-analysis of the APBS has established that the construct measured by the APBS is not very stable and may change over time [75]. Even the reliability estimates decrease with more specific samples (e.g., psychologists, clergy, nurses, and public school teachers) [75]. In terms of its factor structure, there is also no full consensus and different models have been tested based on the best fit with the areas studied, from a one-factor model, which has conceived of prosociality as a global behavioral trend [79], to those that have tested the multidimensional nature of the scale. With respect to these, Luengo Kanacri et al. [70] compared several models discussed in the literature. The one with the best fit had a bifactor approach, which included a general latent factor (prosociality) and two specific factors (prosocial actions and prosocial feelings) [70]. The four theoretical dimensions were considered but regrouping occurred in only one of those referring to the behavioral component, leaving another to represent the empathic dimension. For their part, the model tested by Mieres-Chacaltana et al., which also had a bifactor approach, considered one general factor (prosociality) and four specific factors (helping, sharing, empathy, and caring) [76], i.e., these factors reflected each of the specific dimensions of the prosociality proposed by the creators of the APBS. Despite the above, the results showed the solid unidimensional character of the scale [76] and also coincided in this respect with those found by the authors of this study [74].

Another observed difference has been in the analytical approaches, represented by the confirmatory factor analysis (CFA) in one case [70] and by exploratory structural equation modeling (ESEM) in the other [76]. This could also explain some of the discrepancies because, even though CFA has mainly been used to examine the factor structure of an instrument, they are frequently restrictive and simplistic [80] since cross-loadings are constrained to zero [81,82]. And this does not correspond to the complex nature of some constructs where the items receive indirect influence from factors to which they do not belong at the theoretical level. In that sense, implementing ESEM on theoretically complex constructs could provide a more realistic point of view and, thus, an understanding of the dimensional nature of the construct [83].

Therefore, given the importance it could have for the context in question, an adequate psychometric study is needed to guarantee the reliability and validity of the APBS. The specific objectives are: (i) to evaluate its factor structure and its level of unidimensionality; (ii) to evaluate its reliability; and (iii) to evaluate its factorial invariance by gender. The following hypotheses were derived from these objectives: (H1) the APBS has a unidimensional structure; (H2) the reliability results of the APBS for Chilean teachers are adequate; and (H3) the results of the models for men and women adjust acceptably, demonstrating factorial invariance in terms of gender.

## 2. Materials and Methods

### 2.1. Participants

The population defined for this study comprised all the teachers who work in the first cycle of primary education in public schools in Chile (*n* = 85,298 teachers). A stratified random sample was selected considering the following strata: region, residence (urban, rural), type of funding, and gender. Stratified probability sampling was used and was multistage, with a reliability of 95%, a sampling error of 2.5%, and a variance *p* = *q* = 0.5 [84]. The final sample comprised 1404 teachers, representing 1.65% of the population, aged between 21 and 70 years (*M* = 41.4 years; SD = 10.8). Of these, 1088 (77.5%) were women, and 316 (22.5%) were men.

### 2.2. Instrument

The scale to measure prosociality in adults (APBS) was used; it was constructed by Caprara et al. [74] and validated in Chile by Mieres-Chacaltana et al. [76]. Each item is linked to five categories on an ordinal scale from Never (1) to Always (5). The items include actions referring to helping (e.g., “I am pleased to help my friends/colleagues in their activities”); sharing (e.g., “I am willing to make my knowledge and abilities available to others”); caring (e.g., “I try to be close to and take care of those who are in need”); and feeling empathic with others (e.g., “I am empathic with those who are in need”). Each item is linked to the theoretical dimensions, which comprise prosocial behavior according to Caprara et al. [74]. The content of the items appears in Table 1.

### 2.3. Procedure

The protocol established in the framework of Fondecyt project 1,210,551 was followed. First, all the school principals were contacted. In addition, so were the mayors and directors of the local education services since in Chile public schools are under the administration of the municipalities and the local education services under the Chilean Ministry of Education. The study was presented to all these actors, and authorization was sought so that the schools under their direction could participate.

Data collection was carried out through a software (Question Pro Online Survey, Seattle, WA: QuestionPro Inc, USA). Also, visits and the application of the instruments were scheduled in the schools themselves to ensure the sample size. To protect the ethical principles of the project, informed consent was obtained from all the participants. The study has the approval of the Scientific Universidad de La Frontera, Chile (Assessment File N°053_21; Research Protocol Page N°019/21).

### 2.4. Analytical Approach

The data analysis process was conducted in stages. In the first stage, the most suitable factor structure for the context studied was determined using the evaluation of models suggested by the literature: exploratory structural equation modeling (ESEM). This analytical option was supported by four foundations. First, ESEM is essentially considered a confirmatory technique [85], but it is more flexible and has fewer identification and specification errors than CFA [85,86]. Second, it provides a more adequate representation of the data in terms of fit, particularly for confirmatory purposes [85,86]. Third, it obtains more exact estimations of the relationships between latent factors [87,88]. Fourth, ESEM models tend to align better with the theoretical representation of the construct the instrument is intended to measure [80]. Given that, in certain cases, the CFA model is also nested within an ESEM model, the two can be compared [85]. If it fits the data better than CFA, the estimation of the factor correlation will likely be substantially less biased in the ESEM model than in the CFA model [85]. Thus, this comparison was applied in those cases where it could be made. In addition to fulfilling an exhaustivity criterion, this made it possible to contrast the levels of closeness or the discrepancy of the models with the theoretical conceptualization that attempts to measure the instrument. Figure 1 shows some of the admissible models according to the literature. Models 1 (one-factor model) and 2 (second-order model) were evaluated with CFA. Models 3b, 4b, and 5b are representations of ESEM models. The comparison described in the preceding lines could be applied in these three cases because an evaluation was also conducted with CFA (models 3a, 4a, and 5a).

The Mplus 8.4 program was used [89,90], complemented by the ESEM Code Generator for Mplus [91]. The evaluation took place using the weighted least squares means and variance adjusted (WLSMV) estimation method [92], which is recommended for analyzing ordinal variables [93] in a wide range of sample sizes [94]. In addition, WLSMV makes no distributive assumptions about the observed variables [95]. Consequently, the robust standard errors of the structural coefficients are more precise than those obtained with MLR and ULSMV in every situation of skewed data [96]. In the case of the ESEM model estimations, target rotation was used, which makes it possible to use this technique in a confirmatory mode, given that it produces the rotated solution nearest to a pre-specified configuration of loads [81]. It provides a more robust model a priori and facilitates the interpretation of the results [85]. The goodness of fit was evaluated using the following indicators: the comparative fit index (CFI), the Tucker–Lewis index (TLI), the root mean square error of approximation (RMSEA), and the standardized root mean square residual (SRMR). This last indicator was used to complement the RMSEA because new studies have indicated that the SRMR surpasses the RMSEA when the evaluated data are categorical [97]. From an interpretative point of view, an adequate fit of the model is assumed when the CFI and TLI present values over 0.90 [98], while for the RMSEA, adequate values are below 0.08 [99,100]. Ideally, the lower value of the 90%CI of the RMSEA should be close to zero (or no more than 0.05), and the upper value should be below 0.08 [101]. In relation to the SRMR, a value below 0.08 is considered a good fit [102], and a value below 0.10 is acceptable [103]. In the case of the compared nested models, a better fit is given when the chi-square difference test is significant (*p* < 0.05). In addition, the differences in the RMSEA and SRMR, on the one hand, and those of the TLI and CFI, on the other, must be higher than 0.015 and 0.01, respectively [101,104].

To evaluate the unidimensionality of the scale under bifactor modeling, the explained common variance (ECV), percentage of uncontaminated correlations (PUC), and percentage of reliable variance (PRV) were used [105,106,107,108]. For interpretative purposes, ECV > 0.70 and PUC > 0.70 indicate that the relative bias will be slight, and the common variation can essentially be considered unidimensional [109]. If the PUC is over 0.80, the ECV values are less important in predicting bias [110]. In relation to the PRV, values over 75% indicate strong evidence for the use of the score from a subscale. Additionally, the omega hierarchical (ωh) and omega hierarchical by dimension (ωhs) were considered [109,110,111]. The first more accurately estimates the strength of a general factor in structural equation models [112,113], and values over 0.75 indicate a predominance of only one general factor [110] and permit a clear contrast to the weight of the specific factors [106]. The ωhs evaluates the variance explained by each specific dimension, controlling for the presence of the general factor, and magnitudes over 0.30 indicate that the specific factor can be interpreted [114]. The reliability score was estimated with Cronbach’s alpha coefficient (α) [115] and the construct reliability with the McDonald’s omega (ω) [116] and H index [117]. H is a measure of construct replicability which “represent[s] the correlation between a factor and an optimally-weighted item composite. Then, high H values (>0.80) suggest a well-defined latent variable” [117] (p. 230).

The measurement of invariance between men and women was determined with the selected model, which was examined using two approaches. The first was a general one, based consecutively and cumulatively on the variation of different parameters [118]: the total configuration equivalence of the ABPS (configurational invariance), the factor loadings (weak invariance), and the thresholds (strong invariance). The degree of invariance was calculated, including the variation in the magnitude of the CFI and RMSEA between the nested models. This way, there is unfavorable evidence for the measurement invariance if ΔCFI < −0.01 and ΔRMSEA ≥ 0.01 [104] or if ΔCFI < −0.002 and ΔRMSEA ≥ 0.007 [119]. The other approach is a specific one based on effect size using the comparison of specific parameters (factor loadings, thresholds, and residuals) associated with the invariance between the groups [120,121], taking Cohen’s indices as the reference [122]. Then, to compare the factor loadings of men and women, the coefficient *q* was used; magnitudes below |0.10| were expected to determine that these were equivalent between the groups. In terms of the thresholds, it was expected that the *d* would be lower than |0.20|, and as far as the difference between residuals was concerned, if the *h* statistic was lower than |0.10|, it would be an indicator of equivalence. Later, based on a sufficient degree of strong invariance, the means between the men and women were compared using an effect size approach using Cohen’s *d* [122], where values over |0.41| indicate significant differences between groups [123].

## 3. Results

Table 2 presents the descriptive indicators of the items. For the females, the lowest mean value is 3.32 (item 11), and the highest is 4.56 (item 3) points; for the males, these values are 3.11 and 4.45 points in the same items, respectively. The lowest standard deviation value is observed in item 3 (Female: 0.70 and Male: 0.75), and the highest is in item 11 (Female: 1.15 and Male: 1.17). The 16 items are slightly skewed, with SSI values below 0.25 [124].

In terms of fit, only the ESEM models presented adequate fits (see Table 3). Of these, the one that fits better to the data than the other theoretical models was the bifactor ESEM model with four specific dimensions, identified as model 5b (see Figure 1 and Table 3). Comparisons between models 3a/3b and 4a/4b are also provided.

The differences registered in ΔCFI, ΔTLI, ΔRMSEA, and ΔSRMR favor the selection of ESEM models 3b and 4b over CFA models 3a and 4a, respectively. Moreover, it is possible to add that in the cases of models 3a and 4a the latent variable covariance matrix (psi) is not positive definite. This situation could indicate a negative variance, a variable residual for a latent variable, a correlation greater than or equal to one between two latent variables, or a linear dependency among more than two latent variables. The comparison between models 5a/5b through indicators was impossible given that model 5a could not be identified.

Table 4 presents the standardized factor loadings and the indices calculated to evaluate the reliability and unidimensionality of the APBS. The loads in the general factor of prosociality presented a range that varied between 0.85 and 0.57, and all were significant. The primary loadings of the specific factors were generally very low and nonsignificant. Both the Cronbach’s alpha coefficient and the omega coefficient presented high values of reliability: 0.932 and 0.968, respectively. The H index of the general factor was 0.956, which means a high correlation between the general factor and the optimally weighted items. This suggests that the general factor is a well-defined latent variable. In contrast, the H index values for the specific factors were well below the cutoff value. All the indicators to evaluate the unidimensionality of the scale exceeded their cutoff scores. The highest ECV value was obtained for the general factor and was 0.857. The PUC value for this scale was 0.8, which was higher than the proposed cutoff score of 0.70 and indicated the unidimensionality of the scale. In addition, the ωh of the general factor was 0.94, showing a high reliability of the factor in revealing the subjects’ position on a single continuum of prosociality measured by all the items. However, the ωhs indices referring to the specific factors reached very low values, which made it impossible to interpret them as useful components for the measurement beyond the general factor. The PRV of the general factor was 99.4%, which is strong evidence for the use of the total score. By contrast, all the PRV values for the subscales were below the 75% limit. This evidence contradicts the use of the scores of some of them.

As far as the analysis of the measurement invariance by gender is concerned, for both the variation of indices of fit (Table 5) and the effect size of individual parameter differences (Table 6), prosociality is measured in an equivalent way between men and women, considering factor loadings (weak invariance), thresholds (strong invariance), and residuals (strict invariance). Later, in terms of the differences between groups, no significant differences were found (*d* = 0.27) between the men (*M* = 64.39) and women (*M* = 67.06).

## 4. Discussion

The general objective of the study was to evaluate the factor structure, internal consistency, factorial reliability, and invariance of the APBS in a sample of Chilean teachers.

First, the findings supported a bifactor model comprising a general factor and four specific latent factors. The general factor explained the similarity of the prosocial tendencies that these particular dimensions share when people express behavioral dispositions on behalf of others [70]. The four specific factors reflected those behaviors and feelings that together make up prosociality according to the theoretical concept held by the creators of the APBS [74]. In contrast with the preceding empirical research, this factor structure agrees with the presented one by Mieres-Chacaltana et al. [76] and differs from the other explored options [70,79]. Consequently, these discrepancies highlight the importance of considering the specific contexts in which prosociality unfolds when studying it [56,72,73].

However, beyond the presence of these facets, the APBS in its totality has been strongly unidimensional. The general factor presented high stability and explained most of the reliable variance of the items [105,106,107,108,109,110,111,112,113]. After examining the effect of the overall dimension, the facets contributed almost nothing to the measurement [114]. This trait agreed with the results yielded by the tests conducted in Italy by the authors of the APBS. In a principal component analysis, Caprara et al. [74] compared the percentages of variance explained by the first and second unrotated components, and the ratio was about 5:1, attesting to the unidimensionality of the APBS. This suggests, on the one hand, that the low variance explained by the facets was largely due to residuals and item specificities. On the other hand, none of these dimensions by themselves make it possible to draw a substantive interpretation that explains the display of prosociality. However, the weakness of its psychometric presence in the APBS does not negate its possible theoretical relevance. Instead, it reinforces the predominant conception of a unidimensional construct [54], albeit a complex one [53], that includes specific dimensions expressed on a continuum. And, although they are distinct constructs, the dimensions of prosociality overlap to a certain degree [69]. Consequently, hypothesis 1 (H1) affirming the unidimensional structure of the APBS is supported.

Second, the results corroborated the general reliability of the instrument for measuring prosociality in a professional teaching context [115,116,117], which supports hypothesis 2 (H2). The need for this evaluation was justified based on what had been reported by the previous research, particularly the assertion regarding the declining reliability estimates of the APBS with samples with certain specificities, like profession [75]. On the other hand, depending on the context, the benefits to others may be of different types and may have different values, hence the need to consider them; not doing so has caused inconsistencies in the theoretical discussions [55]. These frames of reference not only differentiate professional spheres, but also demarcate boundaries between those who practice a given profession and those who train for its future practice [78].

Third, the factorial invariance of the APBS was examined based on gender. The results showed that the APBS measures prosociality in an equivalent way between men and women, with both the variation of indices of fit provided by the general approach [104,118,119] and the specific one that measured the effect size of the individual parameter differences [120,121,122]. This study considered factor loadings (weak invariance), thresholds (strong invariance), and residuals (strict invariance), which supports hypothesis 3 (H3). In addition, it coincides with the results of the authors of the APBS since in the analysis of item slope parameters no differences were found between men and women [74]. This is to say, “the 16 items of the prosocialness scale have an equal capacity to discriminate among male and female adults with different levels of prosocialness” [74] (p. 86–87). On the other hand, the results suggest that the different manifestations of prosocial behaviors and feelings considered by the APBS would be equally present in women and men since no significant differences were found [123], even with the possibility of an important socializing effect by the environment [77]. This does indeed mark a difference with the aforementioned study [74] and others that show differences in favor of women [57,58,59,60,63]. Its explanation may lie in socialization patterns [64] associated with emerged stereotypes in the culture [61] and may be established mainly during adolescence [62]. Given its relevance due to its possible replicability [61] and impact on the school trajectory [65], new studies would be required to investigate its implications; in particular, it should be viewed from the perspective of the influences and/or effects that teachers, both men and women, could exert with higher or lower levels of prosociality. Thus, it is legitimate to ask whether these are behavioral manifestations adapted to fit a given context (e.g., teaching involves supporting the student in achieving the learning objectives) or whether they surface genuinely as part of the individual’s personality.

This study contributes to the literature on prosociality in two important ways. First, there is the validation of the instrument with participants belonging to a particular professional sphere. This agrees with the recommendations to evaluate the operationalization of the construct in a specific cultural, social, and professional framework [56,72,73], particularly one where prosociality becomes a relevant job skill [72,73] and—in the specific case of the context studied—authenticated by regulatory requirements [51,52]. Although possibly comparable in different degrees and forms of operationalization, the actions of helping [36,37], sharing [38], caring [39,40,41], and feeling empathy [32,43,44,45,46,47,48] constitute expressions of prosociality in various teaching scenarios, and it is displayed on multiple levels [1]. One is the micro-intrapersonal, which refers to dispositions and tendencies; another is the meso-intrapersonal, concerning behaviors directed to a person or small groups [1]. It is possible to associate both with teaching suitability [20] and the configuration of socioemotional competences that facilitate and/or make possible multiple benefits for children and adolescent students [7,21,22,23,24,25,26,27,28,29,30,31]. A third level, macro in nature, distinguishes behaviors displayed by and directed toward beneficiaries on a large scale, as well as toward group processes [1]. Placing the construct in this perspective makes it possible to determine the dimensions of the critical relevance it acquires for schools in the 21st century [2,3]. The positive or advantageous aspects generated by deploying prosociality in schools benefit their members [5,8,9,10,11,12,13,14,15,16,17,18,19]; in addition, they are projected to society [4,7,8,9,10,20]. Therefore, given its social and research relevance, the APBS is a contribution because it constitutes an easily administered and psychometrically solid measuring instrument. Its application can contribute to the gathering of information to detect individual differences in prosociality among teachers and can associate it with other constructs that could be distinguished. In short, it can generate knowledge as an input to guide processes of initial training and/or teaching improvement in the socioemotional dimension.

Another contribution of the present study has a methodological scope as it illustrated the applicability of ESEM as an option to traditional CFA approaches in assessing the factorial validity of the APBS. The CFA models applied here did not make it possible to adequately describe the data or provide the fits recommended by the literature [97,98,99,100,101,102,103]. Prosociality is a theoretically complex construct for which CFA models have been restrictive and simplistic [80], given that the items are restricted to loading only on their respective subscales. In other words, they do not establish the flexibility to estimate the crossed loads because they are constrained to zero [81,82]. The compared results [101,104] supported the rationale for choosing an ESEM analysis. First, it could be used as a confirmatory technique [81,85] with greater flexibility and fewer identification and specification errors than CFA [85,86]. Second, it provided better fits [85,86]. Third, it obtained more accurate estimations of the relations among the latent factors [87,88], in this case, among the specific facets and between these and the general factor. Fourth, it represented in psychometric terms a better theoretical representation of prosociality [80].

This investigation is not without limitations. First of all, given the focus of the project linked to this study, only the population of primary school teachers in Chile was considered. Therefore, generalizing the findings to teachers at other education levels must be carried out with caution and requires further research. A second limitation lies in the self-reported nature of the data, which is why memory bias could have affected the responses. In the same way, social desirability could have altered them since it is to be expected that people overestimate their prosociality. Despite the potential mitigating effect of the guarantee of anonymity, it would be advisable to establish measures and controls on social desirability. If the voluntary nature of participation is added in, then it is likely that there has been greater participation by people with high levels of prosocial dispositions. On the other hand, the very condition of self-reporting had an impact on the homogenizing nature of the scale responses, which do not allow the specificities of social phenomena, which are always territorially and historically situated and need to be taken into account. Consequently, future research should investigate multimethod approaches and, among other things, determine their level of agreement with self-report-based studies. A third limitation is given by the cross-section of the study, which does not allow the possible variations of prosociality over time to be established, particularly if the information that reports the low stability of the construct measured by the APBS over time is taken into consideration [75]. In this sense, future research can propose longitudinal studies that, among other objectives, study the invariance of the APBS based on the temporal factor. A fourth limitation is the consideration of gender as a dichotomous variable. As the previous investigation reported differences in the prosociality of men and women, it would be interesting to explore its relationships with other gender options and the factors that would explain them, as would its impact on the school from the perspective of the teacher’s daily work. Finally, it was not possible to perform invariance according to the amount of teaching experience due to the absence of an objective criterion to delimit groups according to teaching experience. Given that professional experience can impact on the processes related to prosociality, this is an important pending task for future research. Additionally, invariance in the measurement of prosocial behavior in Hispanic countries is an equally important task.

## 5. Conclusions

In relation to the purpose and objective defined in this research, it may be concluded that the APBS has shown adequate psychometric properties for the context studied. Therefore, it is an adequate tool for conducting studies that seek to measure prosociality in teachers, taking into account the precautions mentioned in the previous section. This would make it possible to generate information and knowledge that would serve as input to provide feedback and strengthen prosocial competences in practicing teachers; these are attributes currently in great demand in the teaching profession. Furthermore, from an applied approach perspective, it is important to consider gender invariance, given the reliability of the instrument to evaluate prosociality in men and women.

Future studies could contribute evidence of transcultural validation of the APBS in other educational contexts to undertake comparative studies.

## Figures and Tables

**Figure 1 behavsci-13-00761-f001:**
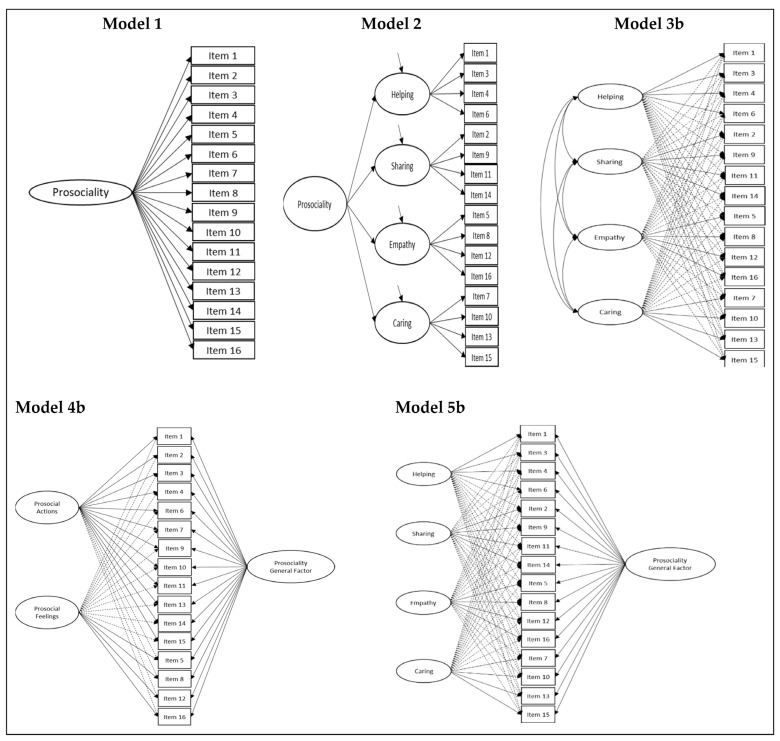
Some of the admissible models for the APBS.

**Table 1 behavsci-13-00761-t001:** Adult Prosocialness Behavior Scale (APBS) [74].

Item	Content
Item 1	I am pleased to help my friends/colleagues in their activities
Item 2	I share the things that I have with my friends
Item 3	I try to help others
Item 4	I am available for volunteer activities to help those who are in need
Item 5	I am empathic with those who are in need
Item 6	I help immediately those who are in need
Item 7	I do what I can to help others avoid getting into trouble
Item 8	I intensely feel what others feel
Item 9	I am willing to make my knowledge and abilities available to others
Item 10	I try to console those who are sad
Item 11	I easily lend money or other things
Item 12	I easily put myself in the shoes of those who are in discomfort
Item 13	I try to be close to and take care of those who are in need
Item 14	I easily share with friends any good opportunity that comes to me
Item 15	I spend time with those friends who feel lonely
Item 16	I immediately sense my friends’ discomfort even when it is not directly communicated to me

**Table 2 behavsci-13-00761-t002:** Descriptive statistics.

	Females (*n* = 1088)					Males (*n* = 316)				
Item	*M*	SD	g1	g2	SSI	M	SD	g1	g2	SSI
1	4.50	0.79	−1.85	3.86	0.155	4.44	0.81	−1.60	2.56	0.135
2	4.54	0.73	−1.90	4.18	0.159	4.35	0.82	−1.31	1.63	0.110
3	4.56	0.70	−1.87	4.37	0.157	4.45	0.75	−1.48	2.64	0.125
4	3.97	1.02	−0.78	−0.09	0.065	3.90	1.08	−0.72	−0.23	0.060
5	4.54	0.72	−1.77	3.58	0.148	4.39	0.82	−1.44	2.04	0.121
6	4.32	0.80	−1.17	1.42	0.098	4.20	0.86	−1.09	1.19	0.091
7	4.13	0.88	−0.89	0.47	0.074	4.00	0.94	−0.88	0.52	0.074
8	3.74	1.00	−0.46	−0.44	0.038	3.36	1.02	−0.25	−0.42	0.021
9	4.51	0.74	−1.69	3.16	0.141	4.51	0.77	−1.87	4.21	0.158
10	4.26	0.89	−1.13	0.74	0.094	3.93	0.97	−0.73	0.14	0.062
11	3.32	1.15	−0.20	−0.75	0.017	3.11	1.17	−0.02	−0.86	0.002
12	4.18	0.88	−1.07	0.98	0.090	3.98	0.90	−0.81	0.56	0.068
13	4.25	0.84	−1.06	0.87	0.089	4.05	0.91	−0.88	0.61	0.074
14	4.25	0.93	−1.23	1.07	0.103	4.16	0.92	−0.88	0.12	0.074
15	3.84	0.98	−0.70	0.14	0.059	3.62	1.02	−0.45	−0.38	0.038
16	4.15	0.90	−1.02	0.77	0.085	3.95	0.98	−0.84	0.217	0.071

Notes: means (M), standard deviations (SD), skewness (g1), kurtosis (g2).

**Table 3 behavsci-13-00761-t003:** Goodness of fit of admissible models for the APBS.

Model	Type of Analysis	χ^2^ (df)	df	CFI	TLI	RMSEA	90%CI RMSEA	SRMR	ΔCFI	ΔTLI	ΔRMSEA	ΔSRMR
1	CFA	2138.924	104	0.918	0.905	0.118	[0.114, 0.122]	0.051				
2	CFA	2043.659	100	0.921	0.906	0.118	[0.113, 0.122]	0.049				
3a	ESEM	1940.704	98	0.925	0.909	0.116	[0.111, 0.120]	0.047				
3b	ESEM	267.787	62	0.992	0.984	0.049	[0.043, 0.055]	0.014	0.067	0.075	−0.067	−0.033
4a	CFA	1810.073	90	0.930	0.907	0.117	[0.112, 0.121]	0.046				
4b	ESEM	578.424	75	0.980	0.967	0.069	[0.064, 0.074]	0.022	0.050	0.060	0.048	−0.024
5a	CFA	The model may not be identified										
5b	ESEM	176.621		0.995	0.988	0.042	[0.036, 0.049]	0.012				

Notes: χ^2^, Ji-square statistic; df, degrees of freedom; CFI, comparative fit index; TLI, Tucker–Lewis index; RMSEA, root mean square error of approximation; SRMR, standardized root mean square residual.

**Table 4 behavsci-13-00761-t004:** Standardized factor loadings resulting from the B-ESEM bifactor model and indicators of unidimensionality and reliability of the APBS.

Items	Theoretical Factor	FG	F1	F2	F3	F 4
Item 1	Helping	0.74 (0.10) *	**0.04 (0.45) ns**	0.25 (0.27) *	−0.10 (0.10) ns	−0.21 (0.02) *
Item 3	Helping	0.82 (0.11) *	**0.09 (0.48) ns**	0.27 (0.26) ns	−0.08 (0.12) ns	−0.22 (0.06) ns
Item 4	Helping	0.68 (0.07) *	**0.32 (0.17) ns**	0.04 (0.03) ns	−0.09 (0.09) ns	−0.01 (0.12) *
Item 6	Helping	0.77 (0.06) *	**0.50 (0.06) ns**	−0.12 (0.12) ns	0.04 (0.09) ns	0.01 (0.16) ns
Item 2	Sharing	0.70 (0.09) *	−0.01 (0.43) ns	**0.51 (0.19) ns**	−0.10 (0.11) ns	−0.10 (0.18) ns
Item 9	Sharing	0.80 (0.04) *	0.05 (0.23) *	**0.03 (0.15) ns**	−0.05 (0.03) ns	−0.08 (0.16) ns
Item 11	Sharing	0.57 (0.03) *	−0.04 (0.03) *	**−0.02 (0.03) ***	0.16 (0.06) ns	0.05 (0.15) ns
Item 14	Sharing	0.67 (0.07) *	−0.03 (0.11) ns	**0.31 (0.21) ***	0.03 (0.08) ns	0.36 (0.05) ns
Item 5	Empathy	0.82 (0.07) *	0.31 (0.17) ns	0.02 (0.03) ns	**0.11 (0.10) ns**	−0.06 (0.12) ns
Item 8	Empathy	0.72 (0.05) *	−0.01 (0.06) *	−0.18 (0.03) *	**0.15 (0.09) ns**	0.06 (0.30) ns
Item 12	Empathy	0.79 (0.04) *	−0.08 (0.06) ns	−0.06 (0.03) ns	**0.59 (0.11) ***	−0.01 (0.16) ns
Item 16	Empathy	0.68 (0.08) *	−0.02 (0.17) ns	0.11 (0.19) *	**0.14 (0.11) ***	0.31 (0.07) ns
Item 7	Caring	0.76 (0.02) *	0.08 (0.04) ns	−0.14 (0.03) *	0.02 (0.04) ns	**0.04 (0.24) ns**
Item 10	Caring	0.81 (0.09) *	−0.19 (0.05) ns	−0.24 (0.11) ns	0.01 (0.16) ns	**0.08 (0.59) ***
Item 13	Caring	0.85 (0.07) *	−0.09 (0.06) ns	−0.09 (0.02) ns	0.15 (0.11) ns	**0.13 (0.34) ***
Item 15	Caring	0.70 (0.14) *	0.00 (0.34) ns	0.16 (0.37) ns	0.04 (0.16) ns	**0.57 (0.05) ***
ECV		**0.938**	0.046	0.059	0.044	0.061
α		**0.932**				
ω		**0.968**				
ωh		**0.945**				
ωhs			0.006	0.005	0.007	0.006
H		**0.989**	0.381	0.440	0.404	0.468
PUC		**0.762**				
PRV		**97.6**	58.3	47.5	71.1	58.4

Note: ECV = explained common variance; α = Cronbach’s alpha coefficient; ω = coefficient omega; ωh = omega hierarchical; ωhs = omega hierarchical subscale; H = H index; PUC = percentage of uncontaminated correlations; PRV = percentage of reliable variance. Values in bold indicate factor loadings in the primary dimension; values in parentheses correspond to the standard error; * = *p* < 0.01; ns = statistically nonsignificant.

**Table 5 behavsci-13-00761-t005:** Measurement invariance of the APBS.

Model	χ^2^	df	CFI	TLI	RMSEA	90%CI	SRMR	CM	ΔCFI	ΔTLI	ΔRMSEA	ΔSRMR
1. Configural invariance	242.421	143	0.996	0.993	0.031	[0.025, 0.038]	0.014					
2. Weak invariance	300.575	198	0.996	0.995	0.027	[0.021, 0.033]	0.018	1	0	0.002	−0.004	0.004
3. Strong invariance	300.575	198	0.996	0.995	0.027	[0.021, 0.033]	0.018	2	0	0	0	0

**Table 6 behavsci-13-00761-t006:** Measurement invariance between men and women: effect size.

	Men						Women						Effect Size					
Item	λ	Θ	τ1	τ2	τ3	τ4	λ	Θ	τ1	τ2	τ3	τ4	λ	τ1	τ2	τ3	τ4	Θ
1	0.78	0.40	−2.49	−1.78	−1.22	−0.23	0.76	0.42	−2.43	−1.82	−1.27	−0.28	−0.01	0.03	−0.02	0.02	0.02	−0.05
2	0.67	0.55	−2.49	−1.81	−1.10	−0.07	0.74	0.46	−2.52	−1.87	−1.27	−0.23	0.04	−0.01	−0.03	0.07	0.07	0.20
3	0.88	0.23	−2.49	−2.08	−1.27	−0.18	0.84	0.30	−2.55	−2.06	−1.40	−0.28	−0.02	−0.03	0.01	0.05	0.04	−0.18
4	0.68	0.53	−1.90	−1.24	−0.42	0.33	0.70	0.51	−2.02	−1.27	−0.49	0.33	0.01	−0.05	−0.01	0.03	0.00	0.04
5	0.84	0.30	−2.49	−1.81	−1.16	−0.14	0.83	0.31	−2.59	−1.91	−1.30	−0.25	0.00	−0.04	−0.04	0.06	0.05	−0.01
6	0.77	0.41	−2.35	−1.70	−0.95	0.20	0.79	0.38	−2.44	−1.80	−1.03	0.12	0.01	−0.04	−0.04	0.03	0.03	0.06
7	0.79	0.38	−2.15	−1.46	−0.68	0.43	0.74	0.45	−2.29	−1.55	−0.75	0.35	−0.03	−0.06	−0.04	0.03	0.03	−0.14
8	0.71	0.50	−1.74	−0.87	0.10	1.11	0.69	0.52	−1.95	−1.03	−0.11	0.89	−0.01	−0.07	−0.06	0.07	0.08	−0.03
9	0.84	0.29	−2.35	−1.96	−1.35	−0.33	0.81	0.34	−2.48	−1.97	−1.35	−0.32	−0.02	−0.06	−0.01	0.00	0.00	−0.10
10	0.74	0.45	−2.08	−1.43	−0.52	0.45	0.78	0.39	−2.28	−1.54	−0.72	0.22	0.02	−0.08	−0.04	0.08	0.09	0.12
11	0.50	0.75	−1.35	−0.46	0.31	1.08	0.56	0.68	−1.43	−0.58	0.22	1.00	0.05	−0.03	−0.05	0.04	0.04	0.14
12	0.80	0.37	−2.24	−1.53	−0.68	0.51	0.76	0.43	−2.30	−1.57	−0.80	0.36	−0.02	−0.02	−0.02	0.04	0.06	−0.13
13	0.83	0.30	−2.24	−1.58	−0.74	0.37	0.82	0.33	−2.42	−1.65	−0.87	0.24	−0.01	−0.07	−0.03	0.05	0.05	−0.05
14	0.72	0.48	−2.49	−1.61	−0.75	0.14	0.64	0.59	−2.38	−1.59	−0.83	0.06	−0.05	0.05	0.01	0.03	0.03	−0.21
15	0.66	0.57	−1.96	−1.04	−0.22	0.83	0.67	0.55	−1.98	−1.18	−0.35	0.72	0.01	−0.01	−0.05	0.05	0.04	0.04
16	0.66	0.56	−2.15	−1.29	−0.65	0.45	0.66	0.57	−2.22	−1.44	−0.74	0.33	0.00	−0.03	−0.06	0.04	0.05	0.00

Note: λ = factor loading; Θ = residual; τ*n* = *n*-th threshold.

## Data Availability

The data that support the findings of this study are not available because they are confidential data.

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
