# Peer review of "Psychometric Validation of the Adult Prosocialness Behavior Scale in a Professional Teaching Context"

_behavsci, 2023, doi:10.3390/bs13090761_

Round 1
Reviewer 1 Report
The article submitted for review is a complete and well-written paper. It contains a large-sample survey, appropriately designed. The conclusions are supported by appropriate statistical apparatus. The only comment I have concerns Hypothesis 1. Hypotheses should be "sharp" - please reformulate the hypothesis "(H1) the APBS has an essentially unidimensional structure"
or clarify what the authors mean when they write "essentially"
Author Response
Thank you very much for your comments. They have helped to improve the article. Attached is a document with responses to your observations.

Reviewer 2 Report
Thank you for the opportunity to review the paper entitled “Psychometric validation of the Adult Prosocialness Behavior 2 Scale in a professional teaching context”. The paper addresses an important topic, and it potentially makes this study interesting. I have found several strengths in the manuscript, especially in the section of statistical analysis. For instance, the large number of participants, the adoption of an ESEM approach (i.e., a very sophisticated approach that allows to go beyond the limits of the classic CFA), and the use of the weighted least squares means and variance adjusted (WLSMV) method to account for the ordinal nature of the observed data, ensure great strength to the entire study. I just highlight some (very) minor issues that I suggest to address prior to accept the manuscript.
- Introduction is clear, and reported references are quite recent. I would like the authors to pay more attention to the part that stresses the need to conduct this study. In other words, the statement (page 3, lines 102-104) “Consequently, prosociality is of critical importance to the teaching profession. Hence the need to construct [71] or evaluate psychometric properties of measuring instruments in this professional area in which prosociality becomes a relevant job skill [72]” appears to me too abrupt.
- Participants come from primary schools; as reported by authors in the discussion section, generalizing the findings to teachers at other education levels is problematic. I suggest to clearly highlight (maybe also in the title) that the study was conducted within the context of primary school.
- The investigation of gender invariance was very important; I also suggest to consider invariance according to the amount of teaching experience (i.e., professional experience can impact on the processes related to prosociality). If it is not possible to add specific analyses at this stage, please consider adding at least one note in this regard in the limits section.
Author Response

(The authors gave the same response as above.)

Reviewer 3 Report
The main research question is intuited, but the authors should make the main research question explicit and answer it in the discussion.
This study contributes to the literature on prosociality in two important ways. First, the validation of the instrument with participants belonging to a specific professional setting. This is consistent with recommendations to evaluate the operationalization of the construct in a specific cultural, social, and professional setting.
Another contribution of the present study has a methodological scope, as it illustrates the applicability of the ESEM as an option to traditional CFA approaches to assess the factorial validity of the APBS.
The validation of this instrument in teachers and in Chile is a novelty and an important contribution.
The authors should address in the introduction some relevant elements such as:
- Commenting on the role of social support and resilience as predictors of prosocial behaviors.
-Empathy as a predictor of prosocial behavior and perceived severity of criminal acts.
- Mention the relationship between prosocial behaviors and addiction problems.
- Prosocial behavior as a protective factor for addiction behaviors.
-Emotional intelligence as a predictor of prosocial behaviors.
-The relationship between criminal thinking and prosocial behavior.
The conclusions are convincing with the evidence and arguments presented.
It would be interesting to propose as future work to achieve invariance in the measurement of prosocial behavior in Hispanic countries, as other authors have achieved with other prosociality instruments.
The references are appropriate and up to date, but it is advisable to check that all are according to the journal's standards.
The tables and figures are very clear.
Author Response

(The authors gave the same response as above.)
